# Identification of Serious Adverse Events in Patients with Traumatic Brain Injuries, from Prehospital Care to Intensive-Care Unit, Using Early Warning Scores

**DOI:** 10.3390/ijerph17051504

**Published:** 2020-02-26

**Authors:** Francisco Martín-Rodríguez, Raúl López-Izquierdo, Alicia Mohedano-Moriano, Begoña Polonio-López, Clara Maestre Miquel, Antonio Viñuela, Carlos Durantez Fernández, Jesús Gómez Correas, Gonçalo Marques, José Luis Martín-Conty

**Affiliations:** 1Advanced Clinical Simulation Center, School of Medicine, Universidad de Valladolid, 47005 Valladolid, Spain; fmartin@saludcastillayleon.es; 2Advanced Life Support Unit, Emergency Medical Services, 47006 Valladolid, Spain; 3Emergency department, Hospital Universitario Rio Hortega, 47012 Valladolid, Spain; 4Faculty of Health Sciences, Universidad de Castilla la Mancha, 45600 Talavera de la Reina, Spain; Alicia.Mohedano@uclm.es (A.M.-M.); Begona.polonio@uclm.es (B.P.-L.); Clara.maestre@uclm.es (C.M.M.); Antonio.vinuela@uclm.es (A.V.); Carlos.durantez@uclm.es (C.D.F.); jesus.gomez@uclm.es (J.G.C.); JoseLuis.MartinConty@uclm.es (J.L.M.-C.); 5Instituto de Telecomunicaciones, Universidad de Beira Interior, 6200-001 Covilhã, Portugal; goncalosantosmarques@gmail.com

**Keywords:** early warning score, emergency medical services, patient safety, medical decision-making, critical care

## Abstract

Traumatic brain injuries are complex situations in which the emergency medical services must quickly determine the risk of deterioration using minimal diagnostic methods. The aim of this study is to analyze whether the use of early warning scores can help with decision-making in these dynamic situations by determining the patients who need the intensive care unit. A prospective, multicentric cohort study without intervention was carried out on traumatic brain injury patients aged over 18 given advanced life support and taken to the hospital. Our study included a total of 209 cases. The total number of intensive-care unit admissions was 50 cases (23.9%). Of the scores analyzed, the National Early Warning Score2 was the best result presented with an area under the curve of 0.888 (0.81–0.94; *p* < 0.001) and an odds ratio of 25.4 (95% confidence interval (CI):11.2–57.5). The use of early warning scores (and specifically National Early Warning Score2) can help the emergency medical services to differentiate traumatic brain injury patients with a high risk of deterioration. The emergency medical services should use the early warning scores routinely in all cases for the early detection of high-risk situations.

## 1. Introduction

The incidence of patients with traumatic brain injury (TBI) in industrialised countries is approximately 200/100,000 inhabitants/year [1], representing one of the most common causes of permanent damage; around 10%–20% of cases are severe and usually require the intensive-care unit (ICU), presenting high mortality [2,3]. The current epidemiological pattern includes traffic accidents, followed by work-related accidents, falls and deliberate self-harm [4].

TBIs are complex situations that can clearly benefit from prehospital care [5]. Advanced trauma life-support actions, developed at the scene or en route, decisively contribute to avoiding brain injuries secondary to hypoxia or hypotension [6,7,8]. When dealing with TBIs in a prehospital setting, the emergency medical services (EMS) must carry out a systematised assessment [9] and must be able to perform complex procedures at the scene, such as advanced airway management, advanced immobilisation techniques, and analgesia procedures [10].

When dealing with patients with a possible TBI, the EMS must make decisions quickly with very little data. Depending on the injury mechanism, an initial systematized assessment and a few additional tests are used to decide, in line with treatment guides, the best treatment strategy for this specific situation. The early warning scores (EWS) are tools that can help the professional to make clinical decisions, predict the risk of deterioration, monitor the evolution of the patient, and facilitate communication between different levels of care, specifically to promote the safety of the patient [11,12].

The use of EWS in patients with major trauma is well-documented in clinical practice [13,14,15]. Similarly, the Glasgow Coma Scale (GCS) is normally used to establish the severity of the TBI [16,17,18]. There are various scores for predicting adverse outcomes after a TBI, but they are designed for use in the emergency department (ED) and ICU [19,20] where advanced diagnostic equipment is available but there are no specific scores to be used in a prehospital setting.

There is a growing interest in being able to detect bed-side those situations that require a rapid clinical response, and different EWS have been developed, used as standard in ED or ICU, but that until now had not been used in the prehospital context, to try to predict or estimate the risk of suffering serious adverse events in a TBI.

The main aim of this study is to compare the diagnostic accuracy of four EWS commonly used in a prehospital setting for the early detection of TBI to specify the need for ICU, and secondarily:To analyze the accuracy of the different scores for a composite outcome of prehospital serious adverse events (*p*SAEs), understood as the need for advanced airway management and/or mean arterial pressure below 70 mmHg at the scene or en route. The standard protocols in the EMS for advanced airway management include orotracheal intubation, use of alternative devices for difficult airway management (laryngeal mask, video-laryngoscope, laryngeal tube, etc.) and mechanical ventilation with transport respirators. The management of hemorrhagic shock includes the use of tourniquets and hemostatic dressings for the management of external hemorrhages, and the use of vacuum splints, vacuum mattress, pelvis immobilizer, serum administration (permissive hypotension), intravenous use of tranexamic acid, fast transportation and hospital notice.To assess the scores for the detection of early mortality (within 48 hours of the index event for any cause in the hospital).

## 2. Materials and Methods

### 2.1. Study Description and Setting

This is a prospective, multicentric cohort study, without intervention of all TBI presenting at hospital via EMS. 

The study was carried out using the Castile and Leon (Spain) EMS system between 1 October 2018 and 30 November 2019 with a reference population of 1,364,952 inhabitants. Seven advanced life-support units (ALS) were selected with attendance at an average of 4 or more incidents daily and their reference hospitals (four tertiary university hospitals and one small general district hospital).

The ALS comprise a physician, an emergency registered nurse and two emergency medical technicians performing standard advanced life support procedures at the scene and en route, according to protocols.

### 2.2. Participants

The sample was selected from among all emergency calls attended consecutively from patients aged over 18 with a prehospital diagnosis of TBI and transported with ALS selected from among all emergency calls from patients over 18 years received consecutively. 

Cases of cardiorespiratory arrest, pregnant women, patients transported on basic life support or discharged in situ were excluded. When the scene was not safe (e.g. attacks), the patient was not assessed for eligibility. Similarly, excluded from the study were those cases in which it was not possible to obtain informed consent or where the vital signs necessary to calculate the scores were not collected in full. Repeat cases were filtered and only the first event for a given patient was considered.

### 2.3. Selection of Early Warning Scores

Scores were selected that are validated, easy-to-apply in a prehospital setting and that use single parameter or multiple parameter systems, or aggregated weighting systems based on determinations of standard vital signs and/or simple clinical observations. Analysis was not performed on scores using complex parameters (e.g., analytical determinations, diuresis, anatomical injuries) such as the Sequential Organ Failure Assessment score [21] or Acute Physiology and Chronic Health Evaluation [22]. Four EWS were selected (see Appendix A
Table A1) with the following validation values:National Early Warning Score 2 (NEWS2) [23], alarm triggers 7 points.Modified Early Warning Score (MEWS) [24], alarm triggers ≥4 points.Triage Early Warning Score (TEWS) [25], alarm triggers 7 points.Modified Rapid Emergency Medicine Score (MREMS) [26], alarm triggers 14 points.

### 2.4. Outcome Measures and Data Collection

For every patient, at the scene or en route, the emergency registered nurse collected the first set of vital signs and clinical observations required to calculate the scores analyzed. The breathing rate was determined by visually observing the rising and falling of the thorax over 30 seconds; if in doubt, the breathing rate was recounted during pulmonary auscultation lasting a full minute, Oxygen saturation, heart rate and systolic arterial pressure were measured using the LifePAK® 15 (Physio-Control, Inc., Redmond, WA, USA) defibrillator/monitor and the temperature with the ThermoScan® PRO 6000 (WelchAllyn, Inc, Skaneateles Falls, NY, USA) tympanic thermometer. In addition, the GCS was performed and the fraction of inspired oxygen (Fi02) was noted down as a percentage along with the patient’s mobility (walking, with help or stretcher/immobile). To determine the blood glucose values, the FreeStyleOptium Neo device (Abbott Laboratories, Chicago, IL, USA) was used with a measuring range of 20–500 mg/dl.

The physician of the ALS collected the patient’s data, arrival, treatment and transport times, the demographic variables (gender, age, trauma type, injury mechanism) and the advanced life-support actions performed (orotracheal intubation, use of opioid analgesics and the use of advanced immobilization and packing techniques).

For the precise linking of the data, between the clinical history from the EMS in paper form and the electronic clinical history from the hospital (JIMENA, SACYL), date, arrival time, first name and surnames, age, gender and, if available, the healthcare card number (personal and non-transferable) were checked. 5 of 6 descriptors were necessary to guarantee the traceability of the data. Once the data had been linked, the identifiers of the patients were anonymized.

Seventy two hours after the index event, an investigator associated with each hospital reviewed the patient’s electronic clinical history to record the hospital outcomes: need for admission, performance of CT scans, surgical interventions, ICU and mortality within 48 hours of the trauma for any reason in the hospital.

The data for all the patients was recorded electronically in a database created for this purpose. The GCS data was categorized in accordance with the Alert-Verbal-Pain-Unresponsive score (AVPU), alert (GCS = 14–15 points), verbal (GCS = 11–13 points), pain (GCS = 9–10 points) and unresponsive (GCS ≤8 points). The FiO2 was categorized by the use of supplemental oxygen on arrival (FiO2 ≥ 0.22 = supplemental oxygen). With the set of vital signs and clinical observations (obtained directly or based on calculation or recoding of other variables) required in the database, all the scores analyzed were calculated as well as the composite outcome of *p*SAEs (need for advanced airway management and/or mean arterial pressure below 70 mmHg on arrival at the ED).

### 2.5. Missing Data

By means of logical, rank and consistency tests, the database was cleaned resulting in a total of 32 variables. Then a full analysis was carried out, variable-by-variable of unknown data, only considered for the analysis of vital signs, in line with previous studies [27,28]. The study variables do not present lost data.

### 2.6. Primary Data Analysis

Three result variables were used, one principal (ICU) and two secondary (*p*SAEs and early mortality within 48 hours) and 11 independent variables (pulse, breathing rate, temperature, systolic blood pressure, oxygen saturation, supplemental oxygen, AVPU scale, mobility level, trauma injuries, GCS and age) to obtain the 4 scores analyzed (NEWS2, MEWS, TEWS and MREMS) and 14 descriptive variables (gender, arrival time, treatment and transportation, type of trauma, injury mechanism, blood sugar, orotracheal intubation, use of major analgesics, advanced immobilization and/or packing, hospital admission, computed tomography (CT) scan and surgical interventions).

All the data were recorded in a XLSTAT® BioMED database for Microsoft Excel® version 14.4.0. (Microsoft Inc., Redmond, WA, USA), and SPSS version 20.0. (IBM, Armonk, NY, USA), and used to perform the subsequent statistical analysis. The data was presented in accordance with the Standards for Reporting Diagnostic Accuracy 2015 statement [29].

We calculated the area under the curve (AUC) of the receiver-operating characteristic (ROC) of the analyzed scores for *p*SAEs, ICU and early mortality. We determined the cut-off point of each scale that offered the highest sensitivity and specificity (Youden’s test), calculating in each case: positive predictive value (PPV), negative predictive value (NPV), positive probability ratio (PPR), negative probability ratio (NPR), positive likelihood ratio (LR+), negative likelihood ratio (LR−), odds ratio (OR) and diagnostic accuracy (DA). By means of nonparametric contrasts, the equality of the AUC obtained was tested.

### 2.7. Ethical Aspects

The Research Ethics Committee of each participating institution approved the study protocol (10/2049, 10/119, MBCA/dgc, PI-18-895 and PI-010-18). All patients (or guardians) signed the informed consent. This study is reported in line with the STrengthening the Reporting of OBservational studies in Epidemiology (STROBE) statement. This study was in accordance with Good Clinical Practice and the Declaration of Helsinki.

## 3. Results

### 3.1. Patients

Our study included a total of 209 cases (Figure 1). The median age was 54 years (25th–75th percentile: 42–70 years), 68 cases (32.5%) were women. The most common injury mechanisms were traffic accidents (93 cases, 44.5%), followed by falls (40 cases, 19.1%) and accidents at work (26 cases, 12.4%). Table 1 shows the characteristics of the population studied. 

The total number of ICU admissions was 50 cases (23.9%), of pSAEs was 29 cases (13.9%), and mortality within 48 hours was 11 cases (5.3%).

The difference between patients admitted to the ICU and those not requiring ICU was significant for all the variables studied with the exception of age, breathing rate, temperature and blood glucose (Table 2). 

Patients admitted to the ICU presented a greater incidence of advanced techniques at the scene (advanced immobilization, orotracheal intubation and/or use of opioid analgesics), greater number of CT scans, surgical interventions and 20% (10 cases) of early mortality (Table 2).

### 3.2. Prognostic Accuracy of the Scores

The diagnostic accuracy of all the scores analyzed for the ICU requirement is shown in Figure 2a. The score with the best performance was the NEWS2 with an AUC of 0.888 (95% confidence interval (CI) 0.81–0.94; *p* < 0.001); the comparison of the curves presented significant differences between the NEWS2 and the other scores (*p* < 0.05), with the exception of the TEWS with which no differences were observed.

In terms of the detection of pSAEs, the score with the best result was the MEWS with an AUC of 0.923 (95% CI: 0.85–0.99; *p* < 0.001) (Figure 2b). In the comparison of the curves, there were no significant differences between the four scores analyzed (*p* > 0.05). To predict early mortality (Figure 2c), the score that gave the best data was also the NEWS2, with an AUC of 0.955 (95% CI: 0.86–1.00; *p* < 0.001) although in this case there were no statistically significant differences when comparing the curves (*p* > 0.05). 

Table 3 shows the different statistics and the cut-off point with better sensitivity and specificity combined (Youden’s test) for the different scores and outcomes studied. In general terms, all the scores analyzed present very high specificity for all the outcomes and specifically excellent sensitivity for the prediction of early mortality. The outcome with the most discreet results is the prediction of the need for ICU where the NEWS2 is the score that behaves better overall with a positive likelihood ratio of 7.35 (95% CI: 4.49–12.04) and negative likelihood ratio of 0.29 (95% CI: 0.18–0.47). 

## 4. Discussion

With this preliminary, prospective, multicentric cohort study, without intervention, of adults with TBI, we assessed the prognostic accuracy of four EWS (NEWS2, MEWS, TEWS, MREMS) to determine the need for ICU and, secondarily, its ability to assess early mortality and pSAEs. In the assessment of the principal variable, it is observed that, although all the scores analysed give good results, it is the NEWS2 and the TEWS that predict admission to ICU better than the MEWS and the MREMS. However, it is apparent that all of the four scores analysed behave similarly to predict early mortality and the pSAEs assessed. 

### 4.1. Comparison with Previous Studies

Works analyzing the use of EWS in a prehospital setting are numerous [14,15,30,31] and more specifically concern the use of EWS in patients with trauma [32,33,34,35]. Studies that specifically analyse scores to predict the risk of deterioration in TBI are scarce. The scales most studied are the Revised Trauma Score [36], the GCS [16,18] and different physiological and analytical parameters [37] or scores on spinal damage [38]. Najafi et al. [39] analyzed the use of different scores to predict the risk of death within 24 hours, including the NEWS, offering our study (although with a mortality outcome within 48 hours) better AUROC, as well as better sensitivity and specificity.

### 4.2. Managing Prehospital Traumatic Brain Injury (TBI)

The identification in a prehospital setting of TBIs with a high risk of deterioration is directly influenced by the early detection of the two physio-pathological situations causing most morbimortality, hypoxia and hypotension [40,41]. Advanced airway management at the scene [42] using orotracheal intubation in those TBIs with a low GCS score, presence of convulsions, agitation, or in case of situations with other associated trauma, is a recognized practice approved by current brain injury treatment guides [43,44]. Similarly, the appropriate handling of hypotension from the initial moments helps to prevent secondary injuries and maintain adequate cerebral perfusion pressure [5,45,46]. The use of advanced measures at the scene for high-risk patients directly influences short-term (admission to ICU and early mortality) and long-term evolution (mitigation of after-effects) [47,48]. 

### 4.3. Early Warning Scores (EWS) and Prehospital TBI

The regular assessment of the risk of deterioration through EWS is a standard procedure in multiple clinical context the use of which is more than contrasted [14,15,28] and which can help to detect patients with TBI and a high risk [49,50]. Determining which patients may be subsidiaries of advanced procedures and rapid transportation to a useful center must be a priority for the EMS [51].

Of all the EWS analyzed in this study, the NEWS2 [23] seems to be the most suitable for use in a prehospital setting. The NEWS2 is validated at prehospital level [52], its use is well-established in many clinical contexts [30,32,39], it only uses standard physiological parameters that can be determined at the scene or en route and may be used in both patients with trauma and those with a medical pathology [53,54]. To be able to calculate the TEWS [25], it is necessary to know whether the patient presents a trauma or not, and the mobility level, and the MREMS [24] uses age to obtain the scoring. The MEWS [26] presented less diagnostic accuracy (statistically significant) for the detection of the need for ICU, and its use is not as established as the NEWS2. Finally, although the four EWS provide very good results, it seems that the NEWS2 has some very useful benefits for the EMS.

### 4.4. Implications

We are faced with a preliminary study that addresses the use of EWS to help with decision-making in TBI in a prehospital setting. All the scores analyzed help with the early detection of pSAEs (orotracheal intubation and hypotension), with the need for admission to ICU or to find out which patients may present early mortality, but they must be interpreted with caution. The use of EWS, specifically the NEWS2, must be a routine procedure for the EMS but should always be guided by an objective and structured full clinical assessment with treatment based on up-to-date guides for each pathology.

As shown in this study, it is clear that the EWS used in a prehospital setting are excellent tools for the early detection of serious adverse events and, therefore, act as predictors of the risk of deterioration, but prospective multicentric studies must be performed with different EMS to be able to externally validate the results presented in this study.

### 4.5. Limitations

The study has various limitations. Firstly, there may be a selection bias since only those TBI cases that, due to opportunity criteria and a specific time window, were assessed and transported using ALS were included, with cases transported using basic life support or other transport vectors not being eligible. This generated a very homogeneous cohort since all the cases were assessed by the physician at the ALS and the purpose of selecting cases was to detect those cases with a risk of deterioration. Secondly, early mortality (within 48 hours) was determined as one of the result outcomes but we are aware that this time limitation in the inclusion of cases is partial and it may be very interesting to study mortality rates after seven or 30 days, or even in the longer term, but the objective of this study is to analyze the influence of the procedures carried out by the EMS at the scene or en route, such that mortality after two days does not seem to be an adequate solution and this is in line with similar publications [27,55,56].

Finally, the data extractors were not blinded. To minimize bias, the result outcomes were clearly defined, pSAEs (orotracheal intubation or mean arterial pressure less than 70 mmHg), admission to ICU and/or early mortality for any reason within the hospital and within 48 hours of the index event. All staff members were trained in correct data collection and the principal investigator made regular visits to each ALS and each hospital to review the cases and resolve any discrepancies. The principal investigator reviewed 50% of the sample. 

## 5. Conclusions

To summarize, the use of EWS (and specifically NEWS2) can help the EMS to differentiate TBI patients with a high risk of deterioration. The EWS are calculated using a few parameters that are easy to obtain, non-invasive and available in a prehospital so the EMS should use the EWS routinely in all cases for the early detection of high-risk situations.

## Figures and Tables

**Figure 1 ijerph-17-01504-f001:**
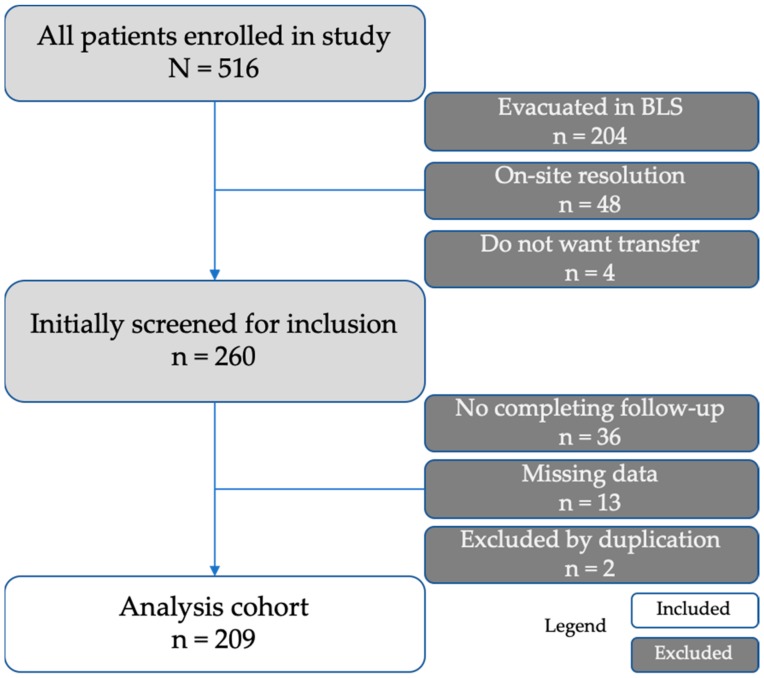
Study participants flowchart. BLS: basic life support

**Figure 2 ijerph-17-01504-f002:**
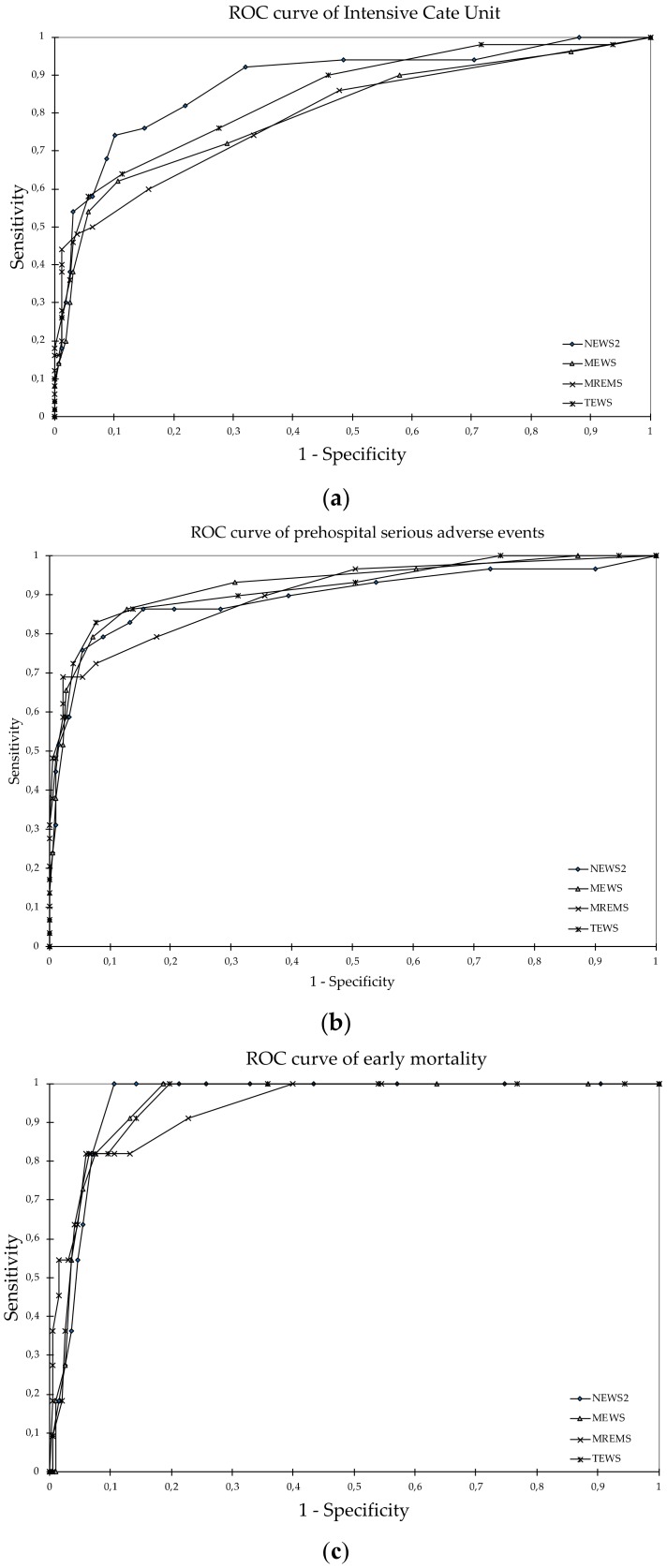
Diagnostic performance curves and areas under the curve: (**a**) Intensive care unit; (**b**) prehospital serious adverse events; (**c**) early mortality. NEWS2: National Early Warning Score 2; MEWS: Modified Early Warning Score; TEWS: Triage Early Warning Score; MREMS: Modified Rapid Emergency Medicine Score

**Table 1 ijerph-17-01504-t001:** Demographic variables (reference rates to intensive care unit inpatients).

		Intensive-Care Unit (ICU)		
Variable ^1^	Total	No	Yes	*p* Value
Number	209	154 (73.7)	50 (23.9)	
Age (years)	54 (42–70)	54 (40–69)	55 (42–74)	0.469
Sex (Female)				0.551
Male	141 (67.5)	109 (68.6)	32 (64.0)	
Female	68 (32.5)	50 (31.4)	18 (36.0)	
Isochronous (minutes)				
Arrival time	9 (7–14)	10 (7–14)	9 (6–15)	0.698
Support time	29 (22–36)	27 (21–34)	35 (25–46)	<0.001
Transfer time	8 (11–16)	10 (8–16)	11 (8–20)	0.236
Injury mechanism				0.756
Traffic accidents	93 (44.5)	71 (44.7)	22 (44.0)	
Pedestrians run over	18 (8.6)	10 (6.3)	8 (16.0)	
Work accidents	26 (12.4)	22 (13.8)	4 (8.0)	
Sports injuries	17 (8.1)	13 (8.2)	4 (8.0)	
Falls	40 (19.1)	34 (21.4)	6 (12.0)	
Precipitates	9 (4.3)	4 (2.5)	5 (10.0)	
Violence	6 (2.9)	5 (3.1)	1 (2.0)	
Injured type				0.330
Penetrating	3 (1.4)	3 (1.9)	0	
Blunt	206 (98.6)	156 (98.1)	50 (100)	

^1^ Values expressed as total number (fraction) and medians [25th percentile–75th percentile] as appropriate.

**Table 2 ijerph-17-01504-t002:** Outcomes of study subjects. Reference rates to intensive-care unit inpatients.

		Intensive-Care Unit			
Variable ^1^	Total	No	Yes	*p* Value	OR ^2^
Age (years)	54 (42–70)	54 (40–69)	55 (42–74)	0.469	1.00 (0.99–1.02)
Initial evaluation					
Pulse (bpm)	85 (74–100)	85 (74–99)	92 (73–112)	0.018	1.01 (1.00–1.03)
BR (bpm)	16 (14–19)	16 (14–19)	18 (12–20)	0.457	1.01 (0.97–1.07)
T (°C)	36.0 (35.8–36.6)	36.0 (36.0–36.7)	36.0 (35.0–36.6)	0.096	0.65 (0.44–0.97)
SBP (mmHg)	130 (118–148)	135 (122–150)	119 (100–140)	0.003	0.97 (0.96–0.99)
SpO2 (%)	96 (94–98)	97 (95–99)	91 (81–96)	<0.001	0.83 (0.77–0.89)
Air oxygen	65 (31.3)	26 (16.4)	39 (78.0)	<0.001	0.05 (0.02–0.12)
GCS (points)	15 (14–15)	15 (15–15)	10 (4–15)	<0.001	0.62 (0.52–0.73)
Mobility level					
Walking	40 (19.1)	38 (23.9)	2 (4.0)	<0.001	2.95 (1.73–5.02)
With help	84 (40.2)	68 (42.8)	16 (32.0)		
Stretcher	85 (40.7)	53 (33.3)	32 (6040)		
Glucose (mg/dL)	120 (101–145)	120 (100–140)	120 (101–166)	0.122	1.00 (0.99–1.01)
Prehospital outcomes					
Immobilization	85 (40.7)	53 (33.3)	32 (64.0)	0.001	3.55 (1.82–6.91)
Intubation	24 (11.5)	1 (0.6)	23 (46.0)	<0.001	134 (17.4–1038)
Opioid analgesics	85 (40.7)	47 (29.6)	38 (76.0)	<0.001	7.54 (3.62–15.7)
MAP < 70 mmHg	14 (6.7)	4 (2.5)	10 (20.0)	<0.001	9.68 (2.88–32.5)
Hospital outcomes					
Inpatients	102 (48.8)	52 (32.7)	50 (100)	<0.001	0 ^3^
CTscan	158 (75.6)	110 (69.2)	48 (96.0)	<0.001	10.6 (2.49–45.7)
Surgery	34 (16.3)	9 (5.7)	25 (50.0)	<0.001	16.6 (6.97–39.8)
Early mortality	11 (5.3)	1 (0.6)	10 (20.0)	<0.001	39.5 (4.91–317)

^1^ Values expressed as total number (fraction) and medians [25th percentile–75th percentile] as appropriate. ^2^ Bracketed numbers indicate 95% confidence interval. ^3^ It is not possible to calculate the Odds ratio (one of the variables has the zero’s value). BR: Breathing rate; T: Temperature; SBP: Systolic blood pressure; MAP: mean arterial pressure; SpO2: Oxygen saturation; GCS: Glasgow Coma Scale.

**Table 3 ijerph-17-01504-t003:** AUROC, cut-off points for combined sensitivity and specificity with best score (Youden’s test) to NEWS2, MEWS, TEWS and MREMS for ICU, pSAEs and early mortality.

Score	NEWS2 ^1^	MEWS ^1^	TEWS ^1^	MREMS ^1^
	**Intensive-care unit (50 cases, 23.9%)**			
Cut-off	7	4	6	7
AUROC	0.88 (0.81–0.94)	0.80 (0.72–0.88)	0.84 (0.77–0.91)	0.79 (0.71–0.87)
p value	< 0.001	< 0.001	< 0.001	< 0.001
Sensitivity	74.0 (60.4–84.1)	62.0 (48.2–74.1)	64.0 (50.1–75.9)	60.0 (46.2–72.8)
Specificity	89.9 (84.3–93.7)	89.3 (83.5–93.2)	88.7 (82.8–92.7)	84.3 (77.8–89.1)
PPV	69.8 (56.5–80.5)	64.6 (50.4–76.6)	64.0 (50.1–75.9)	54.5 (41.5–67.0)
NPV	91.7 (86.3–95.1)	88.2 (82.3–92.3)	88.7 (82.8–92.7)	87.0 (80.8–91.4)
Likelihood ratio +	7.3 (4.5–12.0)	5.80(3.5–9.5)	5.6 (3.5–9.1)	3.8 (2.5–5.8)
Likelihood ratio −	0.3 (0.2–0.5)	0.43(0.3–0.6)	0.4 (0.3–0.6)	0.5 (0.3–0.7)
Odds ratio	25.4 (11.2–57.5)	13.6 (6.4–29.1)	13.9 (6.5–29.7)	8.0 (3.9–16.3)
DA	86.1 (80.8–1)	82.8 (77.1–87.3)	82.8 (77.1–87.3)	78.5 (72.4–83.5)
	**Prehospital serious adverse events (29 cases, 13.9%)**			
Cut-off	7	4	7	10
AUROC	0.89 (0.81–0.97)	0.92 (0.85–0.99)	0.91 (0.84–0.98)	0.90 (0.82–0.97)
p value	< 0.001	< 0.001	< 0.001	< 0.001
Sensitivity	86.2 (69.4–94.5)	86.2 (69.4–94.5)	82.8 (65.5–92.4)	69.0 (50.8–82.7)
Specificity	84.4 (78.4–89.0)	87.2 (81.6–91.3)	92.2 (87.4–95.3)	97.8 (94.4–99.1)
PPV	47.2 (34.4–60.3)	52.1 (38.3–65.5)	63.2 (47.2–76.6)	83.3 (64.1–93.3)
NPV	97.4 (93.6–99.0)	97.5 (93.8–99.0)	97.1 (93.3–98.7)	95.1 (91.0–97.4)
Likelihood ratio +	5.5 (3.8–8.0)	6.7 (4.5–10.1)	10.6 (6.2–18.1)	31.0 (11.4–84.3)
Likelihood ratio −	0.2(0.0–0.4)	0.1 (0.0–0.4)	0.2(0.1–0.4)	0.3 (0.2–0.5)
Odds ratio	33.9 (10.9–105)	42.6 (13.6–133)	56.91(18.8–172)	97.78(23.5–346)
DA	84.7 (79.2–88.9)	87.1 (81.9–91.0)	90.9 (86.2–94.1)	93.8 (89.7–96.3)
	**Early mortality (11 cases, 5.3%)**			
Cut-off	10	4	6	12
AUROC	0.95 (0.86–1.00)	0.95 (0.86–1.00)	0.94 (0.85–1.00)	0.93 (0.83–1.00)
p value	< 0.001	< 0.001	< 0.001	< 0.001
Sensitivity	100 (74.1–100)	100 (74.1–100)	100 (74.1–100)	81.8 (52.3–94.9)
Specificity	89.4 (84.3–93.0)	81.3 (75.3–86.1)	80.3 (74.2–85.2)	93.9 (89.7–96.5)
PPV	34.4 (20.4–51.7)	22.9 (13.3–36.5)	22.0 (12.8–35.2)	42.9 (24.5–63.5)
NPV	100 (97.9–100)	100 (97.7–100)	100 (97.6–100)	98.9 (96.2–99.7)
Likelihood ratio +	9.4 (6.3–14.1)	5.3 (4.0–7.1)	5.1 (3.8–6.7)	13.5 (7.3–24.9)
Likelihood ratio −	0 ^2^	0 ^2^	0 ^2^	0.2 (0.0–0.7)
Odds ratio	0 ^3^	0 ^3^	0 ^3^	69.7 (13.5–359)
DA	90.0 (85.1–93.3)	82.3 (76.6–86.9)	81.3 (75.5–86.0)	93.3 (89.1–96.0)

^1^ Bracketed numbers indicate 95% confidence interval. ^2^ It is not possible to calculate the negative likelihood ratio (one of the variables has the value zero). ^3^ It is not possible to calculate the odds ratio (one of the variables has the value zero). NEWS2: National Early Warning Score 2; MEWS: Modified Early Warning Score; TEWS: Triage Early Warning Score; MREMS: Modified Rapid Emergency Medicine Score; AUROC: area under the receiver operating characteristics; PPV: positive predictive value: NPV: negative predictive value; DA: diagnostic accuracy.

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
