# Peer review of "Identification of Serious Adverse Events in Patients with Traumatic Brain Injuries, from Prehospital Care to Intensive-Care Unit, Using Early Warning Scores"

_ijerph, 2020, doi:10.3390/ijerph17051504_

Round 1
Reviewer 1 Report
Thank you for the invitation to review the article entitled “Identification of serious adverse events in patients with traumatic brain-injured, from pre-hospital care 3 to intensive care unit, using early warning scores” (ijerph-709990). The study aimed to analyze whether the use of early warning scores can help with decision-making in these dynamic situations by determining the patients who need the intensive care unit. The study could possibly provide a prehospital triage tool for patients with traumatic brain injury.
My comments are listed below:
- Minor spelling and grammar errors should be corrected.
For example, “pre-hospital” should be “prehospital”.
line 43, “the Emergency Medical Services (EMS) must carry out…” should be “the emergency medical services (EMS) must carry out…”.
Line 56, “the emergency department (ED) and (ICU)…” should be “the emergency department (ED) and intensive care units (ICU)…”.
English editing throughout the manuscript is suggested.
- The presentation of data should be uniformed throughout the manuscript. For example, some data in Table 3 should be rounded off to the first decimal place.
- Some data in Table A1 is not presented correctly; please correct.
- Line 60-62, the authors defined prehospital serious adverse events (pSAEs) as the need for advanced airway management and/or mean arterial pressure below 70 mmHg before arrival at the ED. Since the pSAEs were important outcome measures in this study, the authors should clearly describe their prehospital protocols regarding airway management and shock treatment.
- Figure 1, the 204 excluded cases should be due to BLS teams, instead of BSL?
Author Response
- Minor spelling and grammar errors should be corrected. For example, “pre-hospital” should be “prehospital”. line 43, “the Emergency Medical Services (EMS) must carry out…” should be “the emergency medical services (EMS) must carry out…”. Line 56, “the emergency department (ED) and (ICU)…” should be “the emergency department (ED) and intensive care units (ICU)…”.
Response point 1: According to the reviewer's recommendations, an orthographic review of the entire document has been carried out.
"Pre-hospital" has been changed to "prehospital", and the numbers indicated by the reviewer (emergency department and intensive care unit) have been lowered.
- The presentation of data should be uniformed throughout the manuscript. For example, some data in Table 3 should be rounded off to the first decimal place.
Response point 2: Table 3 has been revised. In the AUROC that there were three decimals, only two have been placed, and all the data has been reviewed by rounding up the rest of the values even as decimal as the reviewer suggests.
- Some data in Table A1 is not presented correctly; please correct.
Response point 3: The table has been revised and the justification of some fields has been corrected
- Line 60-62, the authors defined prehospital serious adverse events (pSAEs) as the need for advanced airway management and/or mean arterial pressure below 70 mmHg before arrival at the ED. Since the pSAEs were important outcome measures in this study, the authors should clearly describe their prehospital protocols regarding airway management and shock treatment.
Response point 4: We understand the reviewer's concern and as a consensus solution if it seems correct we will introduce the following clarification:
We have replaced "before arrival at the ED" with "On the scene or on the road".
We have also defined advanced airway management: "The standard protocols in the EMS for advanced airway management include orotracheal intubation, use of alternative devices for difficult airway management (laryngeal mask, video-laryngoscope, laryngeal tube, etc.) and Mechanical ventilation with transport respirators. The management of hemorrhagic shock includes the use of tourniquets and hemostatic dressings for the management of external hemorrhages, and the use of vacuum splints, vacuum mattress, pelvis immobilizer, serum administration (permissive hypotension), intravenous use of tranexamic acid, fast transportation and hospital notice.”
- Figure 1, the 204 excluded cases should be due to BLS teams, instead of BSL?
Response point 5: It is a mistake, Figure 1 has been corrected

Reviewer 2 Report
This paper is a cohort study to analyse whether the use of early warning scores (EWS) can help with decision-making in these dynamic situations by determining the patients who need the intensive care unit (ICU) with 209 cases.
The article provides relevant evidence on the factors explored, through a rigorous and consistent methodology.
The study has been authorized by an ethical research committee
The results are consistent and are presented clearly and providing significance. The discussion is adequate and allows the transferability of the results.
Some comments are suggested:
- In the abstract should avoid the use of abbreviations.
- The use of MeSH descriptors is recommended as keywords.
- Although the introduction section is adequate and supported in the literature, it would be necessary to expand that section, emphasizing studies that have been relevant in the use of EWS.
- It should be explained what they mean by preliminary study.
- All ethical and legal information provided in the Study Description and Setting section must be included in a specific section on ethical considerations.
- In the section of participants, reference should be made to the type of sampling used.
- In the Selection of Early Warning Scores section, the validation values of the selected scales must be indicated
- On page 112 "medical doctor" is indicated, does that mean that the physicians who participated in the study had to be PhD?
Author Response
- In the abstract should avoid the use of abbreviations.
Response point 1: We have removed all abbreviations from the abstract.
- The use of MeSH descriptors is recommended as keywords.
Response point 2: The keywords have been reviewed and decuardo to the MeSH have been changed to the following: "Early Warning Score; Emergency Medical Services; Patient Safety; Medical Decision-Making; Critical Care".
- Although the introduction section is adequate and supported in the literature, it would be necessary to expand that section, emphasizing studies that have been relevant in the use of EWS.
Response point 3: To clarify this section we have introduced the following sentence at the end of the introduction and before the objectives of the study:
“There is a growing interest in being able to detect bed-side those situations that require a rapid clinical response, and different EWS have been developed, used as standard in ED or ICU, but that until now had not been used in the prehospital context, to try to predict or estimate the risk of suffering serious adverse events in a TBI.
- It should be explained what they mean by preliminary study.
Response point 4: We share the reviewer's doubts. We think that it is a preliminary study because of the small number of cases we have, although it is true that they are in line with similar publications. As a consensus solution, if the reviewer agrees, the preliminary word of the methodology can be withdrawn, and in that way there would be no doubt.
- All ethical and legal information provided in the Study Description and Setting section must be included in a specific section on ethical considerations.
Response point 5: A new subsection has been created in the methodology section called “Ethical aspects” and the information has been moved to this new subsection.
- In the section of participants, reference should be made to the type of sampling used.
Response point 6: The following clarification has been introduced regarding sample selection:
"The sample was selected from among all emergency calls attended consecutively from patients aged over 18 with a prehospital diagnosis of TBI and transported with ALS selected from among all emergency calls of patients over 18 years received consecutively"
- In the Selection of Early Warning Scores section, the validation values of the selected scales must be indicated.
Response point 7: We have inserted the data requested by the reviewer:
Four EWS were selected (see Table A1) with the following validation values:
- National Early Warning Score 2 (NEWS2) [23], alarm triggers 7 points.
- Modified Early Warning Score (MEWS) [24], alarm triggers ≥4 points.
- Triage Early Warning Score (TEWS) [25], alarm triggers 7 points.
- Modified Rapid Emergency Medicine Score (MREMS) [26], alarm triggers 14 points.
- On page 112 "medical doctor" is indicated, does that mean that the physicians who participated in the study had to be PhD
Response point 8: The reviewer is absolutely right, we have replaced medical doctor with
physician
